# Rectification for Stitched Images with Deformable Meshes and Residual Networks

Yingbo Fan [ID], Shanjun Mao *, Mei Li [ID], Zheng Wu [ID], Jitong Kang and Ben Li

Institute of Remote Sensing and Geographic Information Systems, Peking University, No. 5 Summer Palace Road, Beijing 100871, China; ybfan@stu.pku.edu.cn (Y.F.); mli@pku.edu.cn (M.L.); zheng_wu@pku.edu.cn (Z.W.); 2101210061@stu.pku.edu.cn (J.K.); benli@pku.edu.cn (B.L.)
* Correspondence: sjmao_pku@163.com

**Abstract:** Image stitching is an important method for digital image processing, which is often prone to the problem of the irregularity of stitched images after stitching. And the traditional image cropping or complementation methods usually lead to a large number of information loss. Therefore, this paper proposes an image rectification method based on deformable mesh and residual network. The method aims to minimize the information loss at the edges of the spliced image and the information loss inside the image. Specifically, the method can select the most suitable mesh shape for residual network regression according to different images. Its loss function includes global loss and local loss, aiming to minimize the loss of image information within the grid and global target. The method in this paper not only greatly reduces the information loss caused by irregular shapes after image stitching, but also adapts to different images with various rigid structures. Meanwhile, its validation on the DIR-D dataset shows that the method outperforms the state-of-the-art methods in image rectification.

**Keywords:** image rectangular; deformable mesh; width residual network; global loss function





## 1. Introduction

With the rapid development of image stitching and image fusion technologies, methods for obtaining multi-view or even global perspectives through multiple single viewpoints have been widely applied in human production and daily life [1–5]. For instance, the extensive use of technologies such as panoramic images, autonomous driving, and virtual reality (VR) enables the precise remote observation of scenes by individuals [6–8]. However, in the process of stitching multiple single-view images, it is necessary to align the overlapping regions of different images by adjusting their positions, angles, and local distortions [9]. This often results in irregular boundaries in non-overlapping regions, making it challenging for individuals to adapt and making them prone to misjudgments when observing panoramic images [10].

Some studies have solved irregular boundary selection by directly using smaller rectangular boxes to crop images [11,12]. However, such methods may result in the loss of a large amount of information, which contradicts the original purpose of image stitching, which aims to expand the field of view [13,14]. Additionally, image completion can be employed to predict missing portions of an image and restore its integrity to some extent [15,16]. Nevertheless, its limitations are evident, particularly in cases where the missing portions contain complex structures or highly personalized information, making it challenging for image completion to accurately predict the missing areas [17–19]. This limitation renders image completion unsuitable for applications in fields with high security requirements, such as autonomous driving and industrial production monitoring [20].

To address the aforementioned challenges, this study proposes a deep learning-based image rectification algorithm named RIS-DMRN (Rectification for Image Stitching with Deformable Mesh and Residual Network). The algorithm defines a deformable target mesh for irregularly stitched images, which can be predicted during model training. The

selection of the deformable mesh shape is based on the judgment of the current image's rigid structure by a convolutional neural network, offering three options: triangle, rectangle, and regular hexagon. Once the mesh shape is determined, the prediction network generates an initial predicted mesh based on the input irregular image and its mask matrix. The training process employs a width residual network to predict the initial mesh by the content-aware processing of irregularly stitched images. Subsequently, the input irregular image, predicted initial mesh, and predefined target mesh are collectively input into the width residual neural network for rectification regression. The loss function of the width residual network comprises local and global parts. The local loss function controls the deformation loss of targets within the mesh, while the global-related loss function helps avoid global information loss during the deformation process. Finally, the image rectangular restoration is achieved through continuous iterative regression using residual neural networks (Figure 1).

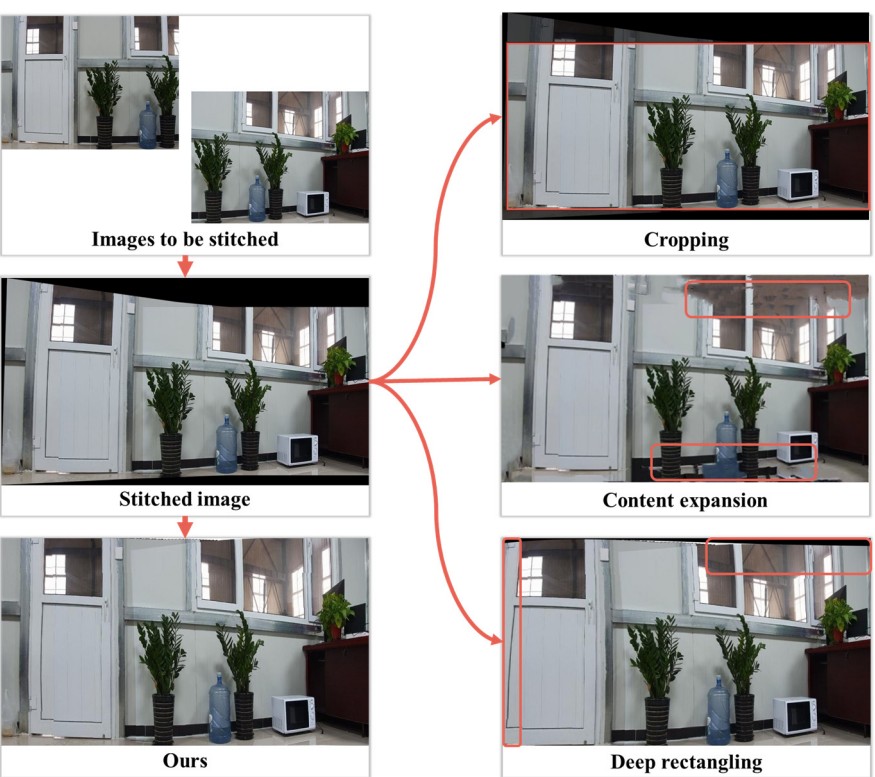

**Figure 1.** Comparison of rectification methods for irregular image stitching.

In response to the current problems of a large loss of edge information and severe deformation of internal details in image rectangles, this paper proposes a method based on variable grids and residual neural networks. The highlights of this study include the following:

- This approach utilizes a deformable mesh as the initial mesh, allowing for more versatile directional movement during mesh restoration and achieving better correction effects for targets within irregular images.
- The introduction of local and global-related loss functions significantly mitigates the drawbacks of traditional methods that focus only on partial regions, enhancing the overall coherence during the deformation recovery process and preserving content more effectively after image rectification.
- In the experiment of randomly parallelizing 300 irregular images on the public dataset DRI-D [21], SSIM and PSNR reached 0.7234 and 22.65, respectively, achieving a relatively accurate level.

## 2. Related Work

Faced with the problems of image rectangles, researchers have conducted some related work and research, including feature matching-based methods, optimization algorithm-based methods, and deep learning methods. However, each method has its own limitations; for example, the feature matching-based method is easily affected by the mismatch and instability of the feature points in the image, resulting in discontinuous or irregularly shaped edges of the spliced image. While the class uses optimization algorithms to adjust the image, it usually needs to define the global or local loss function of the spliced image and use optimization algorithms to minimize the loss function so as to obtain smoother and more regular image edges. However, since these methods require complex optimization calculations on the image and may be affected by local optimal solutions, their effectiveness may be limited when dealing with large-scale images or complex scenes.

For example, in terms of feature matching, Zhu et al. [22] proposed adjusting the stitched image by computing a perspective transformation matrix to make it closer to a rectangular shape. However, this method often relies on the estimation of the geometric structures in specific regions of the image, such as lines or corners. Some approaches suggest transforming local quadrilateral mesh regions on the stitched image to make the overall image more rectangular [23–25]. Building upon the aforementioned research, He et al. [26] proposed optimizing the preservation of line meshes and deforming the rigid structures within the mesh. Li et al. [27] improved the preservation term from line meshes to geodesic lines. However, the applicability of this method is restricted due to the common occurrence of curved ground lines in panoramic images. Some researchers introduced Seam Carving, an algorithmic approach that alters the size of an image by carving or inserting pixels in different parts of the image, thereby transforming irregular images into rectangular forms [28–32]. Meanwhile, Lang et al. [21] proposed DRIS (Deep Rectangling for Image Stitching), employing a residual progressive regression strategy for fully convolutional network prediction of mesh deformations. Based on the predicted mesh, irregular images are corrected. This method partially addresses the challenges of flexible structural distortions for image rectification and computational acceleration. Moreover, the approach utilizes a residual progressive regression strategy for fully convolutional network prediction of mesh deformations and subsequent correction of irregular images. However, DRIS still faces certain challenges. For instance, its loss function focuses solely on the situation within the initial mesh, without considering global information for further adjustments. This limitation results in deformation errors in panoramic information. Additionally, the method concentrates on horizontal and vertical objectives within the mesh, making it prone to deformation errors when correcting targets in other scale directions [33,34]. Currently, there is relatively limited research on image rectification, and achieving image rectification while ensuring minimal loss of information remains a challenging task [35–38].

## 3. Materials and Methods

This paper proposes a deformable mesh structure for the initial prediction of irregular images, enhancing its adaptability in various spatial scene structures. In light of this mesh structure, the paper establishes two methods for mesh application. One approach involves predicting the rigid structure of the input image through a simple convolutional neural network. Based on the prediction results, the most suitable mesh shape for the image is selected (Figure 2a). Subsequently, the input image and the chosen mesh shape are input into a width residual network for initializing mesh prediction. Finally, the predicted initial mesh and the input irregular image are jointly used for image rectification regression, resulting in the output image.

Another option is to input the input image and predefined target meshes for all shapes into a width residual neural network to generate initial mesh predictions (Figure 2b). Subsequently, image rectification regression is performed with the input image. The optimization is then based on the regression loss of the rectified image, selecting the one with the minimum information loss as the final output image.

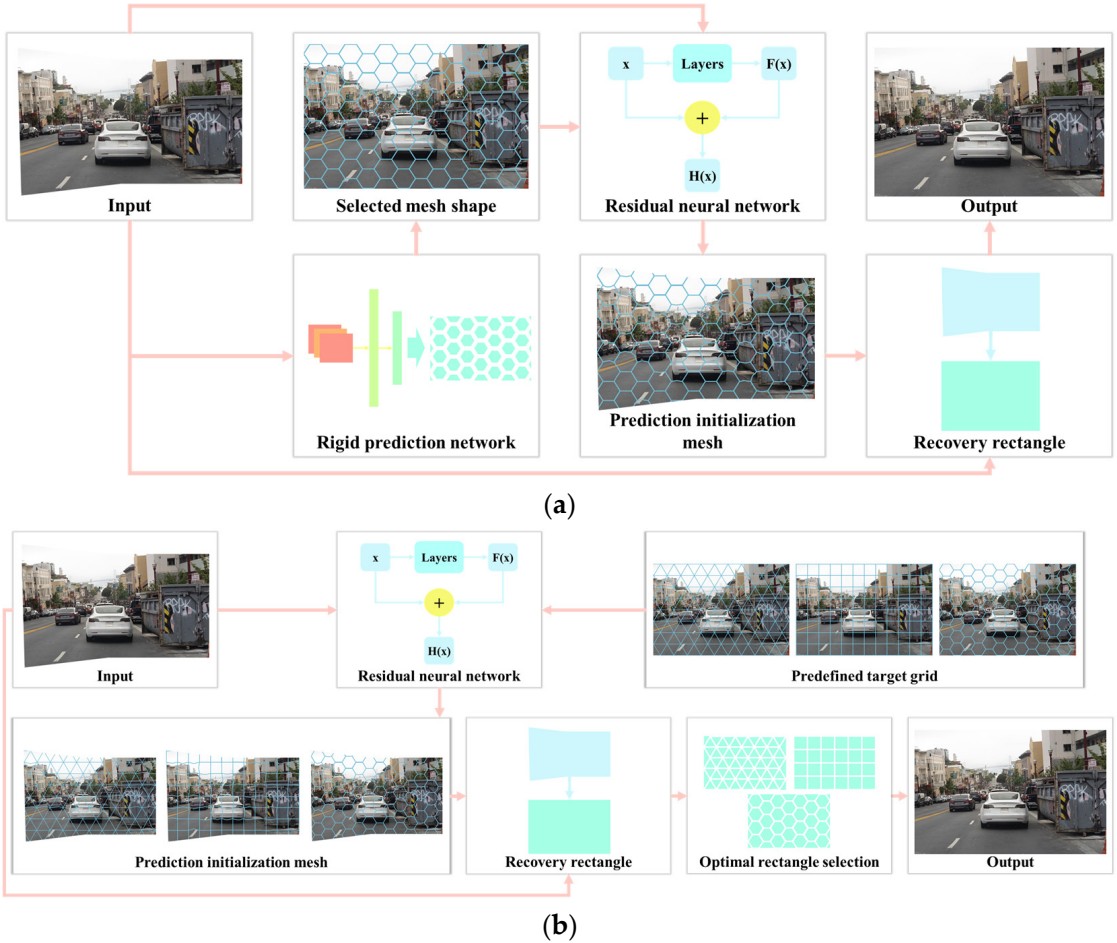

**Figure 2.** Image rectification with different mesh shape selection modes. (**a**) Selecting a mesh shape based on the image. Predicting the most suitable mesh shape for an input image through a convolutional neural network. Subsequently, the predicted initial mesh based on this shape and the predefined mesh are jointly input into a residual network for image rectification. (**b**) Selecting all mesh shapes for loss comparisons. Applying all mesh shapes to the input image and jointly inputting them into a residual network for rectification. The rectified image with the lowest loss is selected as the final output.

A model of image rectification algorithms based on deformable meshes and residual networks is illustrated in Algorithm 1, which consists of two strategies to choose from, namely, the image-based and loss-based strategies. The image-based strategy selects the grid shape that best suits the current image based on the number and distribution of rigid structures within the image. After that, the selected mesh shape and the input image are fed into the residual neural network to obtain the initialized prediction mesh. The initialized prediction grid and the input image are then subjected to rectangle regression together until it meets the accuracy requirements; otherwise, the network continues to be trained. The loss-based strategy, on the other hand, inputs the input image directly into the residual network to obtain initialized prediction grids for the three meshes. Then, after the rectangle regression of the three initialized prediction grids along with the input image, the rectangle image with the lowest loss is selected as the final output.

---

**Algorithm 1** RIS-DMRN algorithm

---

**Input:** Irregular images: *I*; deformable target mesh: *triangle(T)*, *rectangle(R)*, and *regular hexagon(H)*;
  mesh selection strategy: image-based, loss-based; neural network training parameters (such
  as learning rate, batch size, etc.)
**Output:** Rectangle images

---

1:   **for** image-based **do**:
2:       Obtain predicted rigid structure after *I* input CNN
3:       Select the optimal mesh (such as *H*) based on the rigid structure
4:       Input *H* and *I* into RNN to obtain the predicted initialization mesh (*PIM*)
5:       **if** significant losses **do**:
6:           Rectangle regression based on *PIM* and *I*
7:       **else do**
8:           Retrain the network
9:   **end for**
10:  **for** loss-based **do**:
11:      *I* Input RNN to obtain the *PIM* of three meshes
12:      Rectangle regression based on PIM and I
13:      Select the best rectangle image based on three losses
14:  **end for**

---

### 3.1. Deformable Mesh

This paper introduces deformable meshes to meet the application demands in different scenarios. Traditional rectangles exhibit weaker generalization capabilities when dealing with complex scenes. Therefore, the paper introduces two additional mesh models: hexagonal and triangular meshes. Hexagonal meshes have more uniform relationships between adjacent pixels, with each hexagon having six neighbors with equal adjacency properties. This provides better spatial consistency for image rectification. For example, during the image interpolation process, hexagonal meshes can offer smoother and more natural transitions. Moreover, hexagonal meshes closely resemble the shapes of many objects and structures found in the natural world, such as beehives and crystal structures [39]. Therefore, they may provide a more natural representation of images related to natural landscapes. Triangular meshes, on the other hand, excel in realistically reconstructing the shapes in images, especially when the images contain curves and surfaces [40]. The use of triangular meshes allows for better adaptation to irregular image regions, enabling more flexible shape approximation and, consequently, a more accurate capture of details in the images.

Simultaneously, this paper proposes two operational modes for the deformable meshes: speed-oriented and quality-oriented. In the speed-oriented mode, the input image undergoes the detection of rigid structures within irregular images using a simple recognition network. Based on the detected structure count and orientation, the mesh shape that best fits the threshold is directly selected. Currently available mesh shapes include triangles, rectangles, hexagons, and more. In the quality-oriented mode, each mesh shape conducts residual regression predictions on the input image, generating a rectified image. Ultimately, the optimal output is selected based on the loss values of the rectified images, choosing the one with a relatively superior rectification effect.

### 3.2. Network Architecture

The network architecture proposed in RIS-DMRN consists of two components (Figure 3): the rigid target recognition network and the width residual regression network [41]. The input comprises irregularly stitched images and their stitching mask matrix. The input image is initially processed by a simple recognition convolutional neural network to detect the quantity and orientation of rigid structures. Based on the detection results, the most suitable mesh shape is chosen for rectification. For instance, if there are predominantly horizontal or vertical rigid structures in the image, a preference is given to selecting a rectangular mesh. In the case of a higher prevalence of curved surfaces or curved structures,

a triangular mesh is chosen. If the quantities of vertical rigid structures and curved structures are comparable, a hexagonal mesh is selected for image rectification.

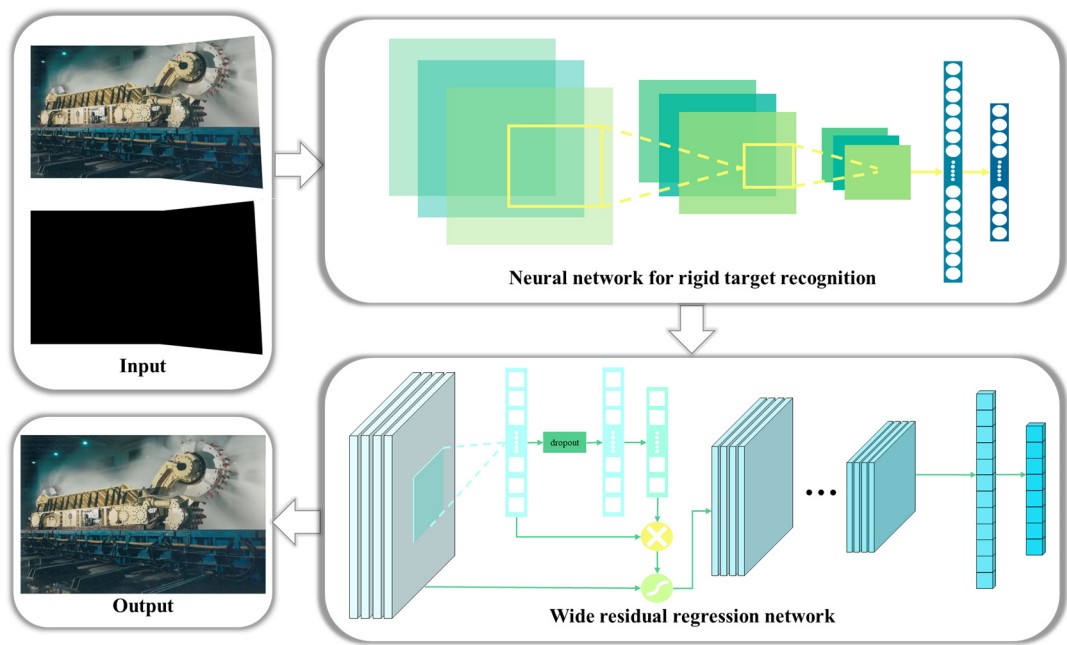

**Figure 3.** Schematic structure of width residual neural network.

The main structure of the rigid structure detection in this paper is a convolutional neural network (Figure 4), with input images resized to a unified $448 \times 448$. The CNN consists of six convolutional blocks, each composed of various combinations of $3 \times 3$ convolutional kernels, $1 \times 1$ convolutional kernels, and $2 \times 2$ max-pooling layers with a stride of 2. After extracting image features into a 1000-dimensional $7 \times 7$ feature vector, a 1000-dimensional vector is generated through average pooling. This vector is then input into Softmax for rigid structure detection. The network also incorporates normalization and dropout operations, although they are not explicitly shown in the diagram.

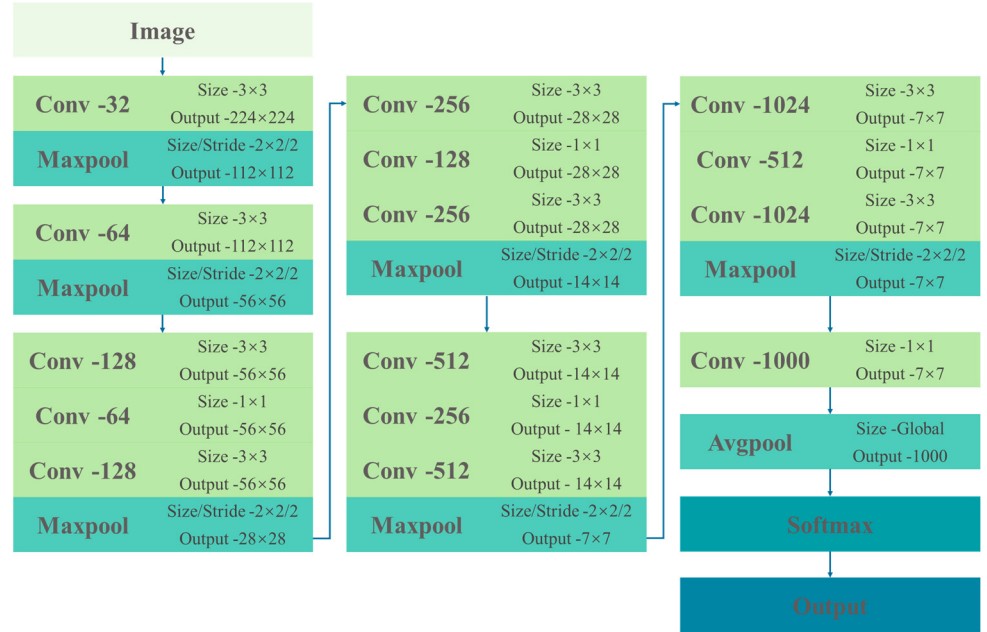

**Figure 4.** Neural networks for target detection in rigid structures.

Once the mesh shape is determined, the irregularly stitched image and its mask matrix are fed into the Wide Residual Neural Network (Wide ResNet) for the rectification process. The choice of Wide ResNet as the recovery prediction network for image rectification is motivated by its ability to enhance feature dimensions in each residual block through increased channel numbers. This augmentation enables the network to capture richer feature representations, playing a crucial role in the recovery of content items after mesh transformation and minimizing the loss in rectified content. Moreover, due to significant internal variations within the meshes during the rectification process, some meshes experience a gradual decrease in gradients during the backward propagation of model training, leading to convergence challenges. The introduction of wider residual blocks in the Wide ResNet facilitates easier gradient flow, mitigating the issue of gradient vanishing during training [42,43].

The architecture of the Wide ResNet employed in RIS-DMRN consists of four residual convolutional blocks followed by an average pooling layer (Table 1). In the network, "k" represents the multiplier for the convolutional kernels in the original module, "N" indicates the number of residual modules in that layer, and "B (3,3)" signifies each residual module consisting of two 3 × 3 convolutional layers. After feature extraction through the residual network, a simple fully convolutional structure is utilized as a mesh motion regressor to predict the horizontal and vertical movements of each vertex based on the regular mesh, facilitating the output of the rectified image.

**Table 1.** Schematic structure of width residual neural network. In the table, "k" represents the multiplier for the convolutional kernels in the original module, "N" indicates the number of residual modules in that layer, and "B (3,3)" signifies each residual module consisting of two 3 × 3 convolutional layers. The image undergoes processing through a mean pooling layer, resulting in the final output image.

| Type | Block Type = B(3,3) | Output |
|---|---|---|
| conv1 | $[3 \times 3, 16]$ | $32 \times 32$ |
| conv2 | $\begin{bmatrix} 3 \times 3, 16 \times k \\ 3 \times 3, 16 \times k \end{bmatrix} \times N$ | $32 \times 32$ |
| conv3 | $\begin{bmatrix} 3 \times 3, 16 \times k \\ 3 \times 3, 16 \times k \end{bmatrix} \times N$ | $16 \times 16$ |
| conv4 | $\begin{bmatrix} 3 \times 3, 16 \times k \\ 3 \times 3, 16 \times k \end{bmatrix} \times N$ | $8 \times 8$ |
| avgpool | $[8 \times 8]$ | $1 \times 1$ |

*3.3. Loss Function*

The loss function of the proposed RIS-DMRN consists of two components: the local loss function and the global loss function. The calculation is formulated as follows in Equation (1):

$$l_{total} = \omega_{local} l_{local} + \omega_{global} l_{global} \tag{1}$$

where $\omega_{local}$ and $\omega_{global}$ represent the weights assigned to the local loss and global loss, respectively. The local loss $l_{local}$ and global loss $l_{global}$ contribute to the overall loss, and the weights control the balance between preserving local details and maintaining global context during the rectification process.

3.3.1. Local Loss

The content loss term in the RIS-DMRN consists of two components: content loss and mesh loss. The content loss term, represented by Equation (2), involves the comparison between the predicted mesh ($m$) applied to the input irregular image ($I_p$) and the warped version of the irregular image ($D(I_p, m)$) using the bending operation. Additionally, the content loss incorporates the difference between the predicted mesh and the ground truth mesh ($T$). The function $C$, denoting the "conv4" convolutional layer in the width recognition network, plays a role in shaping the content loss term. This formulation aims to ensure that

the rectified image aligns closely with both the original irregular content and the ground truth mesh structure.

$$l_{content} = \left\| T - D(I_p, m) \right\|_2 + \left\| C(T) - C(D(I_p, m)) \right\|_2 \tag{2}$$

For the mesh loss term in RIS-DMRN, the formula can be expressed as Equation (3):

$$l_{mesh} = \sum_{i,j} \left\| W(I_p, m_{i,j}) - W(I_p, T_{i,j}) \right\| \tag{3}$$

where $l_{mesh}$ represents the mesh loss term, $i$ and $j$ are indices within the mesh, $m_{i,j}$ is the predicted mesh by the model, $T_{i,j}$ is the ground truth label mesh, and $W$ is the mesh generation function. This loss term aims to encourage the model to better learn and preserve the mesh structure of the image by comparing the differences between the predicted mesh by the model and the true label mesh.

### 3.3.2. Global Loss

The global loss term in RIS-DMRN proposed in this paper consists of two components: global structural loss term and boundary loss term, expressed as shown in Equation (4):

$$l_{global} = l_{ms} + l_{border} \tag{4}$$

The computation of the global structural loss term is expressed as Equation (5), where $I_p^c$ represents the irregular image cropped based on the mask matrix, $\mu_{I_p^c}$ is the mean of $I_p^c$, $\mu_T$ is the mean of $T$, $\sigma_{I_p^c}^2$ is the variance of $I_p^c$, $\sigma_T^2$ is the variance of $T$, and $\sigma_{I_p^c T}$ is the covariance between $I_p^c$ and $T$. Constants $c_1$ and $c_2$ are constants used to stabilize the formula.

$$l_{ms}(I_p^c, T) = \frac{(2\mu_{I_p^c}\mu_T + c_1)(2\sigma_{I_p^c T} + c_2)}{(\mu_{I_p^c}^2 + \mu_T^2 + c_1)(\sigma_{I_p^c}^2 + \sigma_T^2 + c_2)} \tag{5}$$

The expression for the boundary loss term is given by Equation (6), where $I_m$ represents the mask matrix of the original irregular image, and E represents the target template of an all-ones matrix. The boundary loss is adjusted based on the 0/1 mask matrix of the irregular stitched image, with an all-ones matrix as the true target, gradually approaching the rectangularization.

$$l_{border} = \left\| E - D(I_m, m) \right) \right\|_2 \tag{6}$$

## 4. Results

The experimental implementation of the RIS-DMRN algorithm in this study was conducted on the following workstation configuration: Processor (CPU): Intel Core i9-13900HX (2.2 GHz, 6 cores, 12 threads), Memory (RAM): 16GB DDR4 2400MHz, Graphics Card (GPU): NVIDIA GeForce GTX 4070 Ti (8GB GDDR5X). The algorithm was implemented using Python 3.6 + TensorFlow 1.13.1 for program design. Due to the limited availability of publicly accessible datasets for image rectification research, this study conducted validation on the DIR-D dataset [21]. Following the consistent approach outlined in the paper [21], RIS-DMRN set the batch size to 8 during the training process, initialized the learning rate to $1 \times 10^{-3}$, and performed exponential decay on the learning rate at every 50 epochs. The parameters $\omega_{local}$ and $\omega_{global}$ were set to 0.7 and 0.3, respectively, aiming to preserve detailed content while simultaneously focusing on the global shape changes. After the experimentation, this combination was found to be the better choice.

### 4.1. Quantitative Comparison of Image Rectification

The algorithm proposed in this paper was primarily tested on 300 samples selected from the DIR-D dataset. The DIR-D dataset consists of 5839 samples for training and 519 samples for testing. The resolution of each image in the dataset is $512 \times 384$, where

each sample is a ternary consisting of a spliced image, a mask, and a rectangular label. The image content covers most of the scenes in human daily life with good generalization and usefulness, and, thus, serves as the experimental dataset for this paper. A quantitative comparison was performed against mainstream rectification methods, and the results are presented in Table 1. The term "Initialization" denotes the initial state of the freshly stitched image without image rectification processing. The quantification metrics include the average values of SSIM, PSNR, and FID within the samples for comparison [44–46].

It is evident that the proposed RIS-DMRN outperforms RPIW [26] and DRIS [21] in all metrics (Table 2). Additionally, it surpasses traditional seam carving and image completion. This superiority is attributed to the configurations in the loss functions of the deep learning rectification algorithm, which includes the design of content loss and mesh loss. These designs minimize the deformation of target content within the image during rectification, resulting in a more effective image rectification. Compared with the three loss terms of boundary, content, and grid adopted by DRIS, the global structure loss term and boundary loss term in RIS-DMRN better preserve global information. Preserving global information is crucial for the algorithm to better understand the contextual relationships of objects in the image. This understanding is vital for interpreting the relative positions, sizes, and interrelationships of objects. Additionally, global information contributes to maintaining consistency between different regions of the image, ensuring that the algorithm produces coherent output throughout the entire image, especially in tasks like image rectification [47–49].

**Table 2.** Quantization comparison of image rectification on DIR-D. Structural Similarity Index (SSIM), Peak Signal-to-Noise Ratio (PSNR), Mean-Square Error (MSE), and Fréchet Inception Distance (FID) are employed to assess image quality from different perspectives. SSIM measures the structural similarity between two images, considering brightness, contrast, and structure. And the SSIM values range from $-1$ to 1, with 1 indicating identical images. PSNR compares original and processed images by measuring signal-to-noise strength. Higher PSNR values in decibels (dB) indicate better image quality. MSE evaluates the similarity between images by calculating the difference between pixels, with lower values indicating more similar images. FID primarily assesses dissimilarity between generated and real images in terms of distribution. Lower FID values indicate greater similarity in latent space. An upward arrow in the table indicates that the larger its value, the higher the image quality, and vice versa.

| Method | SSIM ↑ | PSNR ↑ | MSE ↓ | FID ↓ |
|---|---|---|---|---|
| Reference | 0.3354 | 11.42 | 3180.97 | 43.57 |
| RPIW [26] | 0.3805 | 15.03 | 2893.10 | 37.51 |
| DRIS [21] | 0.7173 | 21.57 | 1796.83 | 21.26 |
| Ours | 0.7234 | 22.65 | 1512.74 | 20.05 |

Simultaneously, by performing comparisons with the method based on the image's minimum energy line and rectangular mesh division adopted by RPIW, it can be observed that using a hexagon as the initial mesh shape yields slightly better results than traditional rectangular meshes (Table 2). This improvement is attributed to hexagons having more rigid directional choices and better shape adjacency relationships. While hexagons may increase the computational time to some extent compared to rectangles, they often produce superior results.

Consistent with the settings in [21], in this study, we also acknowledge that there may be differences in quantitative measurements when objects undergo slight positional variations in the generated rectangular results. Although the visual perception may still appear very natural in such cases, it could weaken the persuasiveness of quantitative experiments. Therefore, in this study, we also incorporate BIQUE [50] and NIQE [51] as "no-reference" evaluation metrics (Table 3). These two evaluation methods are no-reference image quality assessment metrics dedicated to quantifying the quality of images without the need for any additional reference data [52,53]. It is noteworthy that RIS-DMRN

produces higher quality results under these blind image quality evaluation metrics. This indicates that the proposed method not only excels in preserving global information but also achieves significant improvements in overall image quality.

**Table 3.** Quantitative comparison of non-referenced assessment indicators. Blind Image Quality Evaluator (BIQUE) and Natural Image Quality Evaluator (NIQE) are "no-reference" metrics designed to assess image quality without an original reference image. BIQUE estimates image quality by considering local contrast, structural information, and global color and brightness variations. NIQE focuses on natural images, assessing quality through the analysis of statistical features such as gradients, luminance, and color distribution. An upward arrow in the table indicates that the larger its value, the higher the image quality, and vice versa.

| Method | BIQUE ↓ | NIQE ↓ |
|--------|---------|--------|
| RPIW [26] | 14.045 | 16.927 |
| DRIS [21] | 13.796 | 16.421 |
| Label | 11.017 | 14.763 |
| Ours | 13.562 | 16.027 |

*4.2. Qualitative Comparison of Image Rectification*

To visually demonstrate the effectiveness of RIS-DMRN in image rectification, we divided the test set into two parts—one with more global contextual information and the other with more local detailed information. The algorithm was tested on both sets, and the results were compared qualitatively (Figure 5). Specifically, the study showcases the effects of different input irregular images, image completion results, RIS-DMRN processed images, and ground truth label images in scenes where global correlations are more prominent, such as natural landscapes.

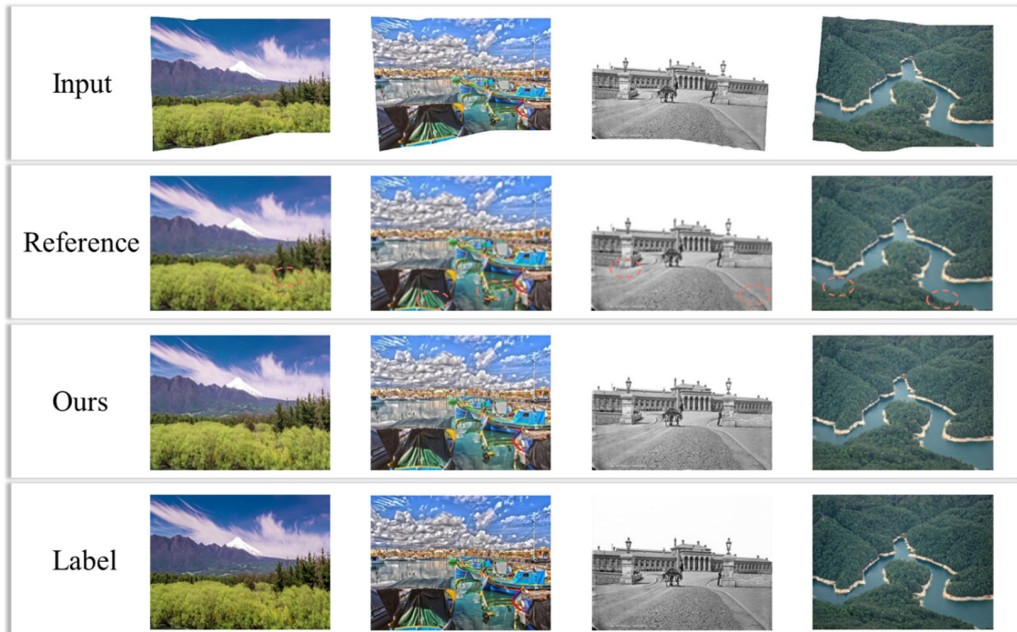

**Figure 5.** Rectification effect of different irregular images under images with more global correlation information. "Reference" in the figure represents the experimental result of image completion.

It is evident that image completion can fully rectify the image into a rectangle (Figure 5), but it relies heavily on pixel-level adjustments based on context, making it overly dependent on surrounding information. This dependency may lead to the inaccurate filling of missing parts, resulting in generated images that appear unrealistic or unnatural, and may

even cause a certain degree of decrease in image clarity [36,54]. In contrast, the rectangular images generated by RIS-DMRN closely resemble the real label images. RIS-DMRN performs well in maintaining the global rigidity or curvature of target objects in the image, attempting to preserve the original appearance without introducing local barrel or pincushion distortions.

In scenarios with dense local information, we compare the results of image completion and RIS-DMRN processing when dealing with irregular images (Figure 6). It is observed that image completion methods often lead to deformations in local rigid structures during the rectification process. Moreover, when addressing the boundaries of missing regions, noticeable boundary effects are common, as image completion methods need to ensure smooth transitions between the filled area and the surrounding region, leading to prominent boundary artifacts [55–57]. In addition, this article also presents other qualitative comparison results, as shown in Appendix A Figures A1 and A2.

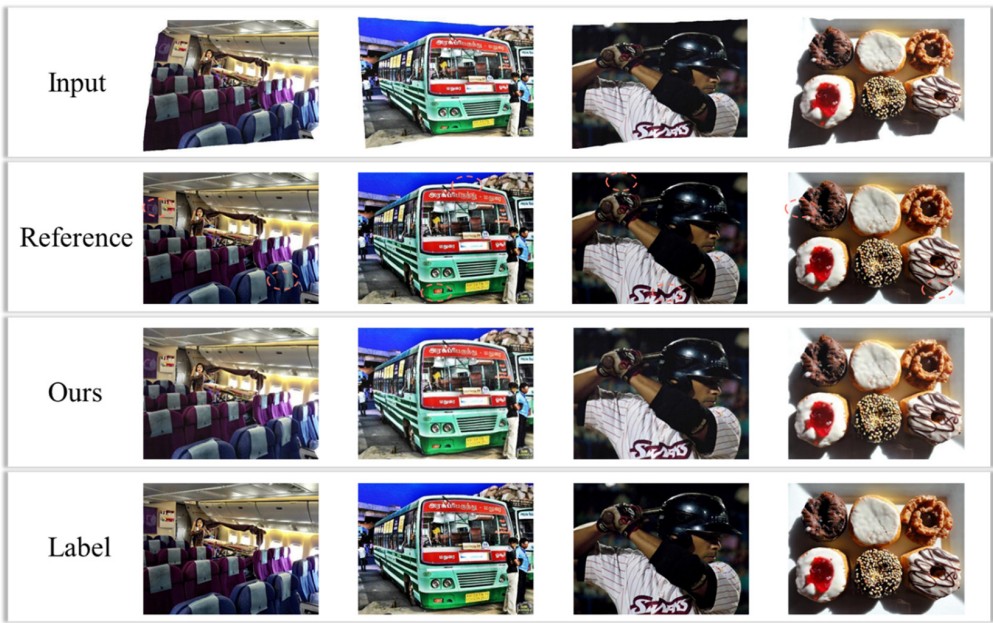

**Figure 6.** Rectification effect of different irregular images under images with more local structural information.

RIS-DMRN, with its finer mesh design, can control the loss and deformation of local information within a certain range. This ensures that local shape changes do not excessively impact the deformation of adjacent meshes, guaranteeing the preservation of local information during the rectification process. Additionally, constraints in different mesh directions in RIS-DMRN allow it to adapt to various deformations of rigid and curved structures, enabling a better fit to the original shape of the image during the deformation process.

### 4.3. Impact of Deformable Mesh and Loss Functions on Image Rectification

In accordance with practical application requirements, we designed three types of deformable meshes—triangle, rectangle, and regular hexagon—for predicting the rectification of irregular images. For the input size of the dataset at $512 \times 384$, a uniform mesh resolution of $16 \times 12$ was employed for rectification prediction. In terms of loss functions, both local and global loss terms were designed for regression prediction. Taking a random selection of 300 images from the test set of the DIR-D dataset as an example, we tested the quantitative metrics for image rectification under different method combinations (Table 4). In the table, "$\mathcal{W}$" indicates the inclusion of the current mesh shape or loss function in the combination.

**Table 4.** Influence of different loss functions and mesh shapes on image rectification. SSIM, PSNR, and FID are utilized as components in various combinations. The symbol "$\mathcal{W}$" denotes the inclusion of the current grid shape or loss function in the combination. The "model" column represents different combinations, where L stands for "Localized loss", G for "Global loss", T for "Triangle", R for "Rectangle", and H for "Hexagon". Various colors are employed in the table for clear correspondence with the combination methods illustrated in Figure 7. An upward arrow in the table indicates that the larger its value, the higher the image quality, and vice versa.

| Loss Function | | Mesh Shape | | | Model | Quantitative Index | | |
|---|---|---|---|---|---|---|---|---|
| Localized Loss | Global Loss | Triangle | Rectangle | Hexagon | Color | SSIM ↑ | PSNR ↑ | FID ↓ |
| | $\mathcal{W}$ | | $\mathcal{W}$ | | **R + G** | 0.4753 | 15.12 | 74.68 |
| $\mathcal{W}$ | | | $\mathcal{W}$ | | **R + L** | 0.6169 | 18.96 | 24.70 |
| $\mathcal{W}$ | $\mathcal{W}$ | $\mathcal{W}$ | | | **T + G + L** | 0.7071 | 20.46 | 22.02 |
| $\mathcal{W}$ | $\mathcal{W}$ | | $\mathcal{W}$ | | **R + G + L** | 0.7126 | 21.05 | 21.74 |
| $\mathcal{W}$ | $\mathcal{W}$ | | | $\mathcal{W}$ | **H + G + L** | 0.7203 | 21.97 | 20.68 |

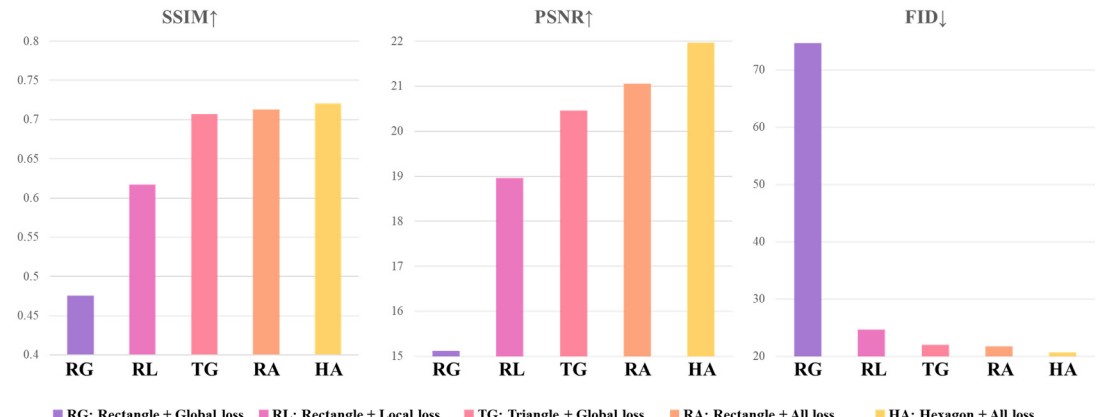

**Figure 7.** Trends of evaluation indexes under different combinations of loss functions and mesh shapes. The chart displays from left to right, with the vertical axis indicating SSIM, PSNR, and FID values, respectively, while the horizontal axis represents different combinations of mesh shapes and loss functions. Higher SSIM and PSNR values indicate better image rectification effects, and a lower FID value suggests superior image rectification performance. An upward arrow in the table indicates that the larger its value, the higher the image quality, and vice versa.

When combining local and global loss terms on the basis of using a rectangular mesh, the overall rectification regression performance is poorer. Comparatively, the absence of the local loss term has a more significant impact on rectification. This is because the global loss term introduces substantial stretching and bending during the regression process, while the local loss term predicts the regression of rigid or curved structures based on the actual content within the mesh. If only the global loss term is used, the rectification may result in severe deformation within local structures, as they are not adequately repaired.

The figures provide a more intuitive sense of the image loss resulting from different combinations, offering a direct visual representation of the changes in evaluation metrics under various combinations (Figure 7). In terms of the overall rectification performance with deformable meshes, it can be observed that the hexagon performs the best, followed by the rectangular mesh, while the triangular mesh exhibits the poorest rectification effect. This is because the hexagonal mesh has better adaptability compared to other meshes; it can effectively cover and adapt to various irregular contours, thereby enhancing the performance of the regression model [58,59].

Simultaneously, the hexagonal mesh can more compactly cover the image area, reducing redundancy. It can decrease edge effects when handling image boundaries, reducing the likelihood of extensive deformation at the image edges and thereby improving repre-

sentation efficiency [60]. In contrast, rectangular meshes may require more mesh points to represent the same image area, resulting in larger input dimensions [61]. Although triangular meshes, compared to rectangular meshes, can better adapt to irregular shapes and exhibit greater flexibility in handling complex image structures [62,63], the special interpolation between triangular meshes and the relationships between neighboring triangles may result in the need for more mesh points to represent the same image area, thus increasing redundancy [64]. Additionally, when dealing with image boundaries, triangular meshes may encounter issues of discontinuity or lack of smoothness at the borders.

## 5. Conclusions and Future Work

This paper proposes a method for rectifying irregularly stitched images using deformable meshes and residual networks. The approach involves predicting initial mesh models for irregular images using three types of shapes: triangles, rectangles, and regular hexagons. The selection of different meshes can be dynamically adjusted based on the requirements of predicting rigid structures or actual image content. The predicted mesh model, predefined mesh model, and irregular input image are jointly input into a width residual network for rectification regression. The loss function comprises local and global loss terms, ensuring that the loss of image information within the mesh and global contextual information is minimized. The final output rectifies the irregularly stitched image into a rectangularized image. The generated rectangularized image not only reduces the information loss in image deformation, but can also be adapted according to different actual input image structures, which further improves the effect of image rectangularization. The image rectangularization has significant advantages in the practical application of image stitching, including simplifying the processing flow, increasing accuracy and stability, and improving the accuracy of stitching results. This enables it to have a wide range of applications in fields such as geographic information systems and medical image processing.

There is still room for further improvement and optimization of the algorithms in this paper. When constructing non-traditional rectangular grids such as triangles or hexagons, the complex calculations require further optimization. For example, calculating the relative positions or distances between neighboring cells may involve more complex geometric operations. Compared to rectangular grids, triangle and hexagonal grids have more complex vertex coordinates and connectivity, and they typically require more storage to represent the same regions. In addition, using neural network structures in deep learning networks for non-rectangular meshes may require more parameters and more complex processing layers, which may lead to slower training and inference. In the future, we will continue to investigate and improve the efficiency of these areas.

**Author Contributions:** Conceptualization, Y.F. and S.M.; Data curation, Y.F.; Formal analysis, Z.W. and B.L.; Funding acquisition, S.M.; Methodology, Y.F.; Project administration, S.M.; Resources, Y.F.; Software, Y.F.; Supervision, S.M.; Validation, Y.F., M.L. and J.K.; Visualization, S.M.; Writing—original draft, Y.F.; Writing—review and editing, Y.F. All authors have read and agreed to the published version of the manuscript.

**Funding:** This research was funded by the National Key R&D Program for the 14th Five-Year Plan (Prevention and Control of Major Natural Disasters and Public Security, Shanjun Mao et al.), grant number SQ2022YFC3000083.

**Institutional Review Board Statement:** Not applicable.

**Informed Consent Statement:** Not applicable.

**Data Availability Statement:** The original contributions presented in the study are included in the article, further inquiries can be directed to the corresponding author.

**Acknowledgments:** We would like to express our gratitude to many colleagues at Beijing LongRuan Technology Co., Ltd., for their extensive assistance in providing data and hardware support for this experiment.

**Conflicts of Interest:** The authors declare no conflicts of interest.

# Appendix A

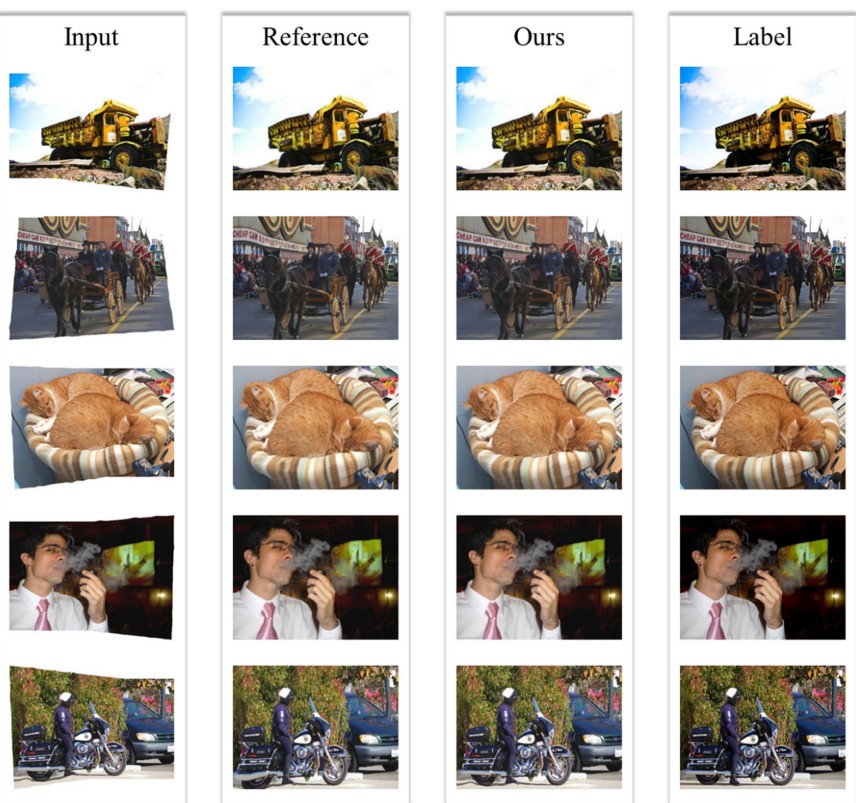

**Figure A1.** Randomly selected stitched images from DIR-D test set for testing the RIS-DMRN.

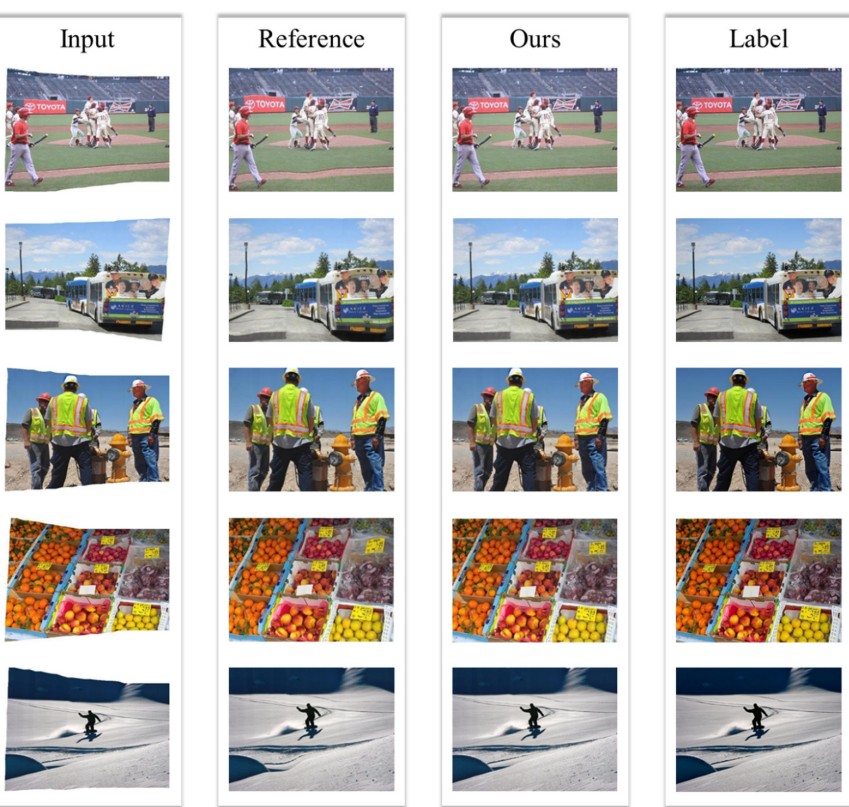

**Figure A2.** Randomly selected stitched images from DIR-D training set for testing the RIS-DMRN.

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
