# Peer review of "Rectification for Stitched Images with Deformable Meshes and Residual Networks"

_applsci, doi:10.3390/app14072821_

Round 1

Reviewer 1 Report

Comments and Suggestions for Authors

Motivation and state-of-the-art are mixed in the Introduction. Authors must separate both concepts. Furthermore, for each work of literature in the state-of-the-art, the authors must indicate the novelty of their work.

There is a significant number of old references. References must be at most 5 years old and must reference works indexed in the Web of Science.

Section 3.2 is confusing, and the authors should consider changing its position to section 3.3.

The conclusions are very unclear and do not adequately include the summary of the system and the results. Future work is not indicated either, except for a brief mention at the end of the Discussion section.

Authors should review figures and tables. For example, Table 4 exceeds the margins, and Figure 7 is difficult to read.

Reviewer 2 Report

Comments and Suggestions for Authors

1. Abstract. 

The authors begin with the description of the problem in order to indicate the work proposal, however, the conceptualization with which they developed the work proposal is confusing; the methodology with which the work was carried out and what was obtained in the work could be described. This would be interesting for the reader, in order for them to continue reading your document. 

2. Introduction. 

The paper talks about the work based on a Residual Neural Network, however in the introduction section it does not talk about the technique to perform the image rectification, in order to understand the authors' proposal.

Results 

Consider augmenting a metric evaluation to assess system performance through structural similarity index, mean square errors or other metrics focused on image rectification. 

It is recommended to update references 

Reviewer 3 Report

Comments and Suggestions for Authors

The authors propose a method for rectifying irregularly images into rectangles using deformable meshes and residual networks, and they use a technique of convolutional neural network to quantify rigid structures of images, such as triangular, rectangular, and hexagonal. And they primarily test on 300 samples selected from the DIR-D dataset. Subsequently, the irregularly image, predefined mesh structure, and predicted mesh structure are inputted into a wide residual neural network for regression function. A quantitative comparison is performed, and the quantification metrics include the average values of SSIM, PSNR, and FID for comparison work. Overall, the assumed benefits of this paper have interesting to some extent for the field of image processing application problems from my viewpoint; however, this paper has some minor remarks. I hope the authors can follow on my comments to radically revise their manuscript for publication in the leading journal of Applied Sciences in MDPI. Thus, by giving the potential practical interest from this paper and the proposed algorithm, I believe this paper has some merits for publication in the stitching image issue. Here I pose some minor drawbacks, as follows.

1.   I suggest the Abstract part should be rewritten in a refined statement and focused on research methodology and empirical results. Now, it is not enough and unclear.

2.   For the comparative experiments, the selection reason for these models and the data sets used should be first defined clearly.

3.   After carefully reviewing the manuscript, I do not see a clear discussion on the important point "their methodology related to earlier works" - do we have anything new or not? Or perhaps, could they tell us what material is new? And why the new material is important? It is useful and helpful to explain them and to tell the truth to the readers for improving the readability of the manuscript.

4.   There are few explanations of the rationale for the proposed algorithm or method designed. Please give the comments.

5.   It seems that there is no section focusing on literature reviews. By presuming that the methods and results represent the literature review, the authors may create some confusion to general readers. I suggest that there is a need for adding a thorough traditional literature review section.

6.    Whether this study has some limitations for their research that can be addressed?

7. The paper seems some theoretical. More potential application results or discussions for practical applications on stitching image issue would enhance the paper.

8.  Also, please make a more explanation for the main results of application level into the Conclusions parts for valuing the study.  Because the contents in Conclusion section are duplicated with beginning of Section 3.1, and it should be revised.

9.  Please check all literature format in References part throughout the text, they should be a consistent format and should follow the style of this Applied Sciences Journal; they exist different formats or something wrong; please revise them. For example, in Line 43: Zhu proposed adjusting the stitched image by …; in Line 49, He proposed optimizing …; in Line 50, Li improved the …; in Line 55, Lang proposed DRIS …

10. Please recheck all the literature in References part; some literature has incomplete or not enough information.

11.  I think that this manuscript needs for proofreading to improve the paper quality.

Comments on the Quality of English Language

Minor editing of English language required

Round 2

Reviewer 1 Report

Comments and Suggestions for Authors

The authors should improve the text in the figures, it is too small and makes it difficult to read.
